# Large Amplitude Iris Fluttering Detected by Consecutive Anterior Segment Optical Coherence Tomography Images in Eyes with Intrascleral Fixation of an Intraocular Lens

**DOI:** 10.3390/jcm11154596

**Published:** 2022-08-06

**Authors:** Makoto Inoue, Takashi Koto, Akito Hirakata

**Affiliations:** Kyorin Eye Center, Kyorin University School of Medicine, 6-20-2 Shinkawa, Mitaka 186-8611, Tokyo, Japan

**Keywords:** intrascleral fixation, intraocular lens, anterior segment optical coherence tomography, iris capture, reverse pupillary block, peripheral iridectomy

## Abstract

Saccadic eye movements induce movements of the aqueous and vitreous humor and iris fluttering. To evaluate iris fluttering during eye movements, anterior segment optical coherence tomography (AS-OCT) was used in 29 eyes with pars plana vitrectomy (PPV) and intrascleral fixation of an intraocular lens (ISF group) and 15 eyes with PPV and an IOL implantation into lens capsular bag (control group). The height of the iris from the iris plane (the line between the anterior chamber angles) was compared every 0.2 s after the eye had moved from a temporal to the primary position (time 0). The height of the nasal iris in the ISF group decreased to −0.68 ± 0.43 mm at 0 s (*p* < 0.001) and returned to −0.06 ± 0.23 mm at 0.2 s. The height of the temporal iris increased to 0.45 ± 0.31 mm at 0 s (*p* < 0.001) and returned to −0.06 ± 0.18 mm at 0.2 s. The height of the nasal iris at 0 s in the ISF group was significantly lower, and that of the temporal iris was significantly higher than the control (−0.05 ± 0.09 mm, 0.03 ± 0.06 mm, *p* < 0.001, respectively). Iris fluttering can act as a check valve for aqueous and vitreous humor movements and can be quantified by consecutive AS-OCT images. Large amplitude iris fluttering in eyes with intrascleral fixation is important because it can lead to a reverse pupillary block.

## 1. Introduction

Intrascleral fixation or ciliary suturing of an intraocular lens (IOLs) has been used on eyes without adequate support by the lens capsule [1,2,3,4,5,6,7,8]. A fluttering of the iris can be seen more frequently in eyes without a lens capsule behind the iris through the slit-lamp microscope when the patient blinks. The flow of the aqueous and vitreous humor through the gap around the optics of the IOL during eye movements causes the iris to oscillate back and forth, i.e., flutter. This is important because excessive iris movements can lead to a pupillary capture by the intraocular lens, causing a reverse pupillary block, one of the major complications after intrascleral fixation and ciliary sulcus fixation of an IOL [9,10,11,12]. Pupillary capture by the IOL can cause an increase in intraocular pressure and mild eye pain and discomfort, called Uveitis-Glaucoma-Hyphema (UGH) syndrome. In addition, the capture by the IOL can deform the pupil, which can reduce visual acuity by decreasing the depth of focus and increasing ocular aberrations.

The eye is filled with fluid, and saccadic eye movements and body motions cause movement of the intraocular fluids, which generates shear stress on the ocular tissues in contact with the fluids [13]. The movement of the aqueous humor and vitreous humor during eye movements have been analyzed with a computational model because it is difficult to observe and measure them in situ in live human eyes [13,14,15,16,17].

The anterior segment optical coherence tomographic (AS-OCT) devices were designed to record cross-sectional images of the anterior segment of the eye [18,19]. The smoothness of the anterior surface of the iris in Fuchs uveitis [20,21], morphological change after laser peripheral iridectomy in eyes with primary angle closure glaucoma and uveitic secondary glaucoma have been evaluated by AS-OCT [22,23]. Swept-source AS-OCT uses a longer wavelength light source of 1310 nm, which enables the scanning of the anterior and posterior surface of an IOL. It also enables high-speed scanning leading to images of the entire anterior chamber angle, and the repetitive imaging of the same section of the anterior chamber will appear similar to a continuous movie scan [18,19]. With the dynamic scanning ability of swept-source AS-OCT, evaluation of iris movement during pupil constriction with light revealed that the iris stiffness was higher in patients with primary angle-closure glaucoma than in healthy controls [24]. Detection of peripheral anterior synechia has been described by corneal deformation using an air-puff dynamic AS-OCT system [25].

We present a dynamic analysis with swept-source AS-OCT, which allowed us to evaluate and quantify the degree of iris fluttering produced by eye movements in eyes with an intrasclerally fixed IOL. The purpose of this study was to determine whether iris flutter immediately after a saccadic eye movement can be detected and quantified. Another purpose was to determine whether the movement of the iris can lead to a pupillary capture by the IOL after intrascleral fixation.

## 2. Materials and Methods

This single-center, observational study was approved by the Institutional Review Committee of the Kyorin University School of Medicine (1507). It adheres to the tenets of the Declaration of Helsinki. All of the patients received a detailed explanation of the surgical and ophthalmic procedures, and all signed an informed consent form. All of the patients consented to our review of their medical records and their anonymized use in medical publications.

### 2.1. Subjects

The findings in 29 eyes of 29 patients after pars plana vitrectomy (PPV) and intrascleral fixation of an IOL (ISF group) were compared to that in 15 eyes of 15 patients after PPV combined with cataract surgery with implantation of an IOL in the lens capsular bag (control group). In the ISF group, the surgery was performed on 21 eyes with IOL dislocation, 4 eyes with lens dislocation, and 4 eyes with aphakia. In the control group, there were 6 eyes with an epiretinal membrane, 6 eyes with a macular hole, 1 eye each with a lamellar macular hole, proliferative diabetic retinopathy, and rhegmatogenous retinal detachment. The eyes with a posterior synechia, impending reverse pupillary block, and intraoperative complications, including posterior capsule rupture during cataract surgery in the control group, were excluded. The age, sex, laterality, axial length, and anterior chamber depth (ACD) were compared between the ISF and the control groups and with and without peripheral iridectomy (PI) in the ISF group.

The axial length was measured with the OA2000 Optical Biometer (TOMEY Corp, Nagoya, Japan). Swept-source AS-OCT (CASIA2, TOMEY Corp, Nagoya, Japan) with a scan rate of 50,000 A-scan per second was used. The ACD was measured from the corneal endothelium to the anterior surface of the IOL along the visual axis with caliper software integrated into the AS-OCT device.

### 2.2. Measurement of Iris Flutter by Anterior Segment Optical Coherence Tomography (AS-OCT)

Cross-sectional images of the anterior segment, including the iris, were obtained by swept-source AS-OCT (CASIA2). Consecutive images were recorded with the movie mode from both groups one month after the surgery. For this, the patient’s head was fixed to the head holders of the AS-OCT device, and the patient was instructed to fixate a target located 30 degrees laterally from the primary visual axis for 1 s and then move the eye back to the primary position and to hold this eye position for 3 s (Figure 1a,b). This sequence was repeated several times, and the sequences with blinking were excluded from the analyses. The timing of the eye movements was controlled by the sound of a metronome with a repetition rate of one second. Video images were recorded with the AS-OCT device at 5 frames/sec. In each sequence, the time when the eye stopped at the primary position was set to 0.

The iris height was defined as the distance to the anterior surface of the middle sector of the iris from the iris plane, which is a reference line connecting the anterior chamber angles (Figure 1c) and measured at the primary position for the temporal and nasal sectors with the caliper function of the ImageJ software (National Institute of Health, Bethesda, MD, USA). The baseline values were measured at the primary position without an eye movement. The iris height was compared between the eyes in the ISF group and the control group. The iris height was also compared with or without PI in the ISF group.

### 2.3. Surgical Procedures

The PPV and all other surgical procedures were performed by two surgeons (TK and MI) with 27-gauge instruments of the Constellation^®^ Vision System (Alcon Laboratories, Fort Worth, TX, USA) under local anesthesia. The surgery of the ISF group was for a dislocated or subluxated crystalline lens or an IOL, or aphakia. For these eyes, the dislocated lens was lifted by suction of a vitreous cutter and then phacoemulsified or extracted. A dislocated acrylic or silicone IOL was cut into 2 or 3 pieces by scissors and then extracted through a transconjunctival single plane corneoscleral incision. A residual posterior hyaloid cortex was made more visible by intravitreal injection of triamcinolone acetonide (MaQaid^®^, Wakamoto Pharmaceutical Co., Ltd., Tokyo, Japan), and the residual posterior hyaloid cortex was removed by suction with a vitreous cutter if it was present. 

The intrascleral fixation was performed as described by Yamane et al. [7,8]. The limbal positions where the haptics of the IOL were extracted were at approximately 2 and 8 o’clock, which were marked with a toric marker. Two angled parallel incisions were made at 2 mm posterior to the limbus at the marked positions by 30-gauge thin-walled needles. A 3-piece IOL with 7 mm diameter optics (NX-70, Santen Pharma, Osaka, Japan) was implanted in the anterior chamber, and both haptics of the IOL were inserted into 30-gauge needles and extracted with the needles. The excess lengths of the haptics were cut, and the ends were flanged with a coagulator (Accu-Temp^®^, Alcon Laboratories, Fort Worth, TX, USA).

The control group underwent PPV combined with cataract surgery and implantation of an IOL in the lens capsular bag before the PPV. Posterior vitreous detachment, membrane peeling, internal limiting peeling with the aid of brilliant blue G, endo-photocoagulation, tamponade by air, or 20% sulfur hexafluoride (SF6) were performed as needed.

### 2.4. Statistical Analyses

The significance of the differences in the baseline characteristics and the degree and time course of the iris fluttering between the 2 groups was determined by the Wilcoxon signed rank tests, Mann–Whitney U tests, or Fisher’s exact probability tests. All statistical analyses were performed using SPSS (version 28.0; IBM, Armonk, New York, NY, USA).

## 3. Results

The differences in age, sex distribution, and axial length between the ISF and the control groups were not significant (Table 1). However, the ACD of the eyes at the baseline in the ISF group was significantly deeper, and the heights of the iris in the temporal and nasal sectors were significantly lower than that of the control group when the eye was in the primary position.

### 3.1. Time Course of Changes in Height of Temporal and Nasal Sectors of Iris after an Eye Movement

At the time when the AS-OCT was used to evaluate iris fluttering, one month after the implantation of the IOL, no intravitreal gas bubbles were seen in any of the eyes that had undergone a gas tamponade in both groups. 

After the eye moved from the lateral to the primary position, the height of the nasal sector of the iris in the ISF group decreased significantly from the baseline at −0.20 ± 0.16 mm to −0.88 ± 0.40 mm at 0 s (*p* < 0.001, Wilcoxon signed-rank test, *p* < 0.01 was taken to be significant with Bonferroni correction, Figure 2 and Figure 3, Appendix A). The height of the nasal iris returned to −0.27 ± 0.29 mm at 0.2 s (*p* = 0.207), to −0.24 ± 0.22 mm at 0.4 s (*p* = 0.163), to −0.22 ± 0.17 mm at 0.6 s (*p* = 0.396), and to −0.21 ± 0.17 mm at 0.8 s (*p* = 0.977) while the eye remained in the primary position. The height of the nasal iris in the control group decreased from the baseline at −0.06 ± 0.17 mm to −0.11 ± 0.19 mm at 0 s (*p* = 0.058) and returned to −0.05 ± 0.16 mm at 0.2 s (*p* = 0.813), to -0.08 ± 0.17 mm at 0.4 s (*p* = 0.041), to −0.06 ± 0.17 mm at 0.6 s (*p* = 0.475), and to −0.06 ± 0.16 mm at 0.8 s (*p* = 0.762) but were not significant (*p* < 0.01 was taken to be significant with Bonferroni correction). 

The amount of height change of the nasal iris from the baseline was significantly greater at 0 s in the ISF group (−0.68 ± 0.43 mm) than in the control group (−0.05 ± 0.09 mm, *p* < 0.001, Figure 2 and Figure 4, Appendix A). However, the amount of height change of the nasal iris was not significantly different between these 2 groups at 0.2 s (*p* = 0.154), 0.4 s (*p* = 0.931), 0.6 s (*p* = 0.832), and 0.8 s (*p* = 0.89). The amount of height change of the nasal iris from the baseline was only significant at 0 s in the ISF group, and this amount of height change of the nasal iris at 0 s was not significant group with age (*p* = 0.321, Spearman partial correlation coefficient), ACD (*p* = 0.446), and axial length (*p* = 0.365). The amount of height change of the nasal iris from the baseline was not significant at 0 s in the control group, and this height change at 0 s was also not significant with age (*p* = 0.745), ACD (*p* = 0.788), and axial length (*p* = 0.176). 

The direction of the changes in the height of the temporal sector of the iris was in the opposite direction from that of the nasal sector. Thus, the height of the temporal iris in the eyes of the ISF group increased significantly from the baseline (−0.23 ± 0.13 mm) to 0.22 ± 0.32 mm at 0 s (*p* < 0.001, Wilcoxon signed-rank test, *p* < 0.01 was taken to be significant with Bonferroni correction), and then returned to −0.29 ± 0.22 mm at 0.2 s (*p* = 0.071), to −0.26 ± 0.18 mm at 0.4 s (*p* = 0.124), to −0.25 ± 0.15 mm at 0.6 s (*p* = 0.008), and to −0.23 ± 0.14 mm at 0.8 s (*p* = 0.483). The height of the temporal iris in the control group decreased from the baseline at −0.08 ± 0.16 mm to −0.05 ± 0.19 mm at 0 s (*p* = 0.091) and returned to −0.07 ± 0.16 mm at 0.2 s (*p* = 0.682), to −0.08 ± 0.14 mm at 0.4 s (*p* = 0.694), to −0.07 ± 0.15 mm at 0.6 s (*p* = 0.027), and to −0.08 ± 0.15 mm at 0.8 s (*p* = 0.079) but were not significant (*p* < 0.01 was taken to be significant with Bonferroni correction). 

The amount of change in the height of the temporal iris from the baseline was significantly greater (0.45 ± 0.31 mm, *p* < 0.001) in the ISF group at 0 s than that in the control group (0.03 ± 0.06 mm). However, the amount of height change of the temporal iris was not significantly different between these 2 groups at 0.2 s (*p* = 0.084), 0.4 s (*p* = 0.244), 0.6 s (*p* = 0.019), and 0.8 s (*p* = 0.089). The amount of height change of the temporal iris from the baseline was only significant at 0 s in the ISF group, and this amount of height change of the temporal iris at 0 s was not significant with age (*p* = 0.386, Spearman partial correlation coefficient), ACD (*p* = 0.482), and axial length (*p* = 0.082). The amount of height change of the temporal iris from the baseline was not significant at 0 s in the control group, and this change at 0 s was significant with axial length (*p* = 0.009) but not with age (*p* = 0.086), and ACD (*p* = 0.889).

### 3.2. Changes in Iris Height in Eyes with and without Peripheral Iridectomy (PI)

The age, sex distribution, axial length, and heights of temporal and nasal sectors of the iris were not significantly different between the eyes with and without PI in the ISF group (Table 2). However, the ACD in eyes without PI was significantly deeper than that in eyes with PI. 

The height of the nasal sector in eyes with PI decreased significantly to −0.88 ± 0.40 mm from the baseline (−0.24 ± 0.13 mm, *p* ≤ 0.001, Wilcoxon signed-rank test, *p* < 0.01 was taken to be significant with Bonferroni correction) at 0 s and returned to −0.32 ± 0.28 mm at 0.2 s (*p* = 0.131), −0.29 ± 0.23 mm at 0.4 s (*p* = 0.117), −0.25 ± 0.17 mm at 0.6 s (*p* = 0.67), and −0.24 ± 0.16 mm at 0.8 s (*p* = 0.981). The height of the nasal sector of the iris in eyes without PI decreased significantly to −0.88 ± 0.42 mm at 0 s from the baseline (−0.14 ± 0.20 mm, *p* = 0.008, Wilcoxon signed-rank test, *p* < 0.01 was taken to be significant with Bonferroni correction). Then, the height of the nasal iris returned to −0.16 ± 0.30 mm at 0.2 s (*p* = 1.0), −0.13 ± 0.17 mm at 0.4 s (*p* = 0.859), −0.15 ± 0.17 mm at 0.6 s (*p* = 0.362), and −0.13 ± 0.17 mm at 0.8 s (*p* = 1.0, Figure 5).

The height of the temporal sector of the iris in the eyes with PI increased to 0.20 ± 0.26 mm at 0 s from the baseline (−0.25 ± 0.11 mm, *p* < 0.001, Wilcoxon signed-rank test, *p* < 0.01 was taken to be significant with Bonferroni correction). The height of the temporal iris then decreased significantly to −0.35 ± 0.22 mm at 0.2 s (*p* = 0.025), −0.32 ± 0.18 mm at 0.4 s (*p* = 0.017), −0.29 ± 0.15 mm at 0.6 s (*p* = 0.015), and returned to −0.24 ± 0.16 mm at 0.8 s (*p* = 0.827, Figure 5). The height of the temporal iris in eyes without PI increased significantly to 0.26 ± 0.43 mm from the baseline (−0.17 ± 0.15 mm, *p* = 0.008, Wilcoxon signed-rank test, *p* < 0.01 was taken to be significant with Bonferroni correction) at 0 s. The height of the temporal iris then returned to −0.15 ± 0.14 mm at 0.2 s (*p* = 0.858), −0.14 ± 0.11 mm at 0.4 s (*p* = 0.477), −0.16 ± 0.12 mm at 0.6 s (*p* = 0.574), and −0.13 ± 0.17 mm at 0.8 s (*p* = 0.249).

The height change of the nasal iris from the baseline in eyes with PI was not significantly different from that of eyes without PI at 0 s (*p* = 0.982), 0.2 s (*p* = 0.444), 0.4 s (*p* = 0.417), 0.6 s (*p* = 0.764), and 0.8 s (*p* = 0.594). The amount of height change of the temporal iris in eyes with PI was not significantly different from that without PI at 0 s (*p* = 0.627), 0.2 s (*p* = 0.153), 0.4 s (*p* = 0.069), 0.6 s (*p* = 0.062), and 0.8 s (*p* = 0.417). 

## 4. Discussion

During saccadic eye movements of normal eyes, the iris, crystalline lens capsule, and anterior vitreous cortex form a barrier that prevents the aqueous humor and vitreous humor from flowing into the other chamber. These barriers then prevent iris flutter. The absence of the vitreous gel after vitrectomy is believed to enhance the flow of vitreous humor into the anterior chamber during eye movements. Silva and associates [13] reported that the computational fluid dynamics of the vitreous humor during saccadic eye movements were different for the gel phase in the vitreous cavity than that with the liquid phase because the inertial effects were more significant with the liquid vitreous. In contrast, the shear stress produced by the gel phase of vitreous humor was more than twice that of the liquid phase. In this study, all eyes were vitrectomized. However, it has been described that vitrectomy without gas tamponade is not associated with changes in anterior chamber morphology evaluated with AS-OCT [26].

After an intrascleral fixation of an IOL without a lens capsular support, the height of the temporal sector of the iris increased, and the nasal sector decreased immediately after the eye moved from a temporal gaze to the primary position. The aqueous humor is more susceptible to inertia moments than the vitreous fluid in vitrectomized eyes during eye movements because the aqueous humor is located further from the center of the eye. When the eye moves from a temporal gaze to the primary position and comes to rest at the primary position, the nasal sector of the iris is pushed posteriorly by the movement of the aqueous humor toward the posterior chamber, and the height of the nasal sector of the iris is pushed downward, i.e., decreased height. At the same time, the temporal sector of the iris is pushed upward and anteriorly toward the anterior chamber as the aqueous humor moves nasally and the vitreous liquid flows into the anterior chamber and the temporal height of the iris increases. 

When the nasal sector of the iris is pushed posteriorly, the pupil margins contact the anterior surface of the IOL, and the flow of the aqueous humor stops as if a valve was closed (attachment of the pupil margin to the IOL). On the other hand, even if the temporal sector of the iris is lifted due to the inflow of vitreous humor into the anterior chamber, the pupil margin does not contact the IOLs. Thus, the flow of the vitreous humor into the anterior chamber on the temporal side is enhanced more than the outflow of aqueous humor on the nasal side as the valve is opened. These movements change the iris height, which then returns to the baseline level after 0.2 s. However, the heights of both the nasal and temporal sectors of the iris remain lower than the baseline and gradually return to the baseline level after 0.4 s. The decreased height of the iris indicates the process of recovery from the influx of vitreous humor into the anterior chamber.

A reverse pupillary block occurs when either side of the iris is pushed posteriorly during eye movements and the pupillary margin does not stay on the anterior surface of IOL but is pushed behind the IOL by the flow of aqueous humor. A PI was thought to be able to prevent a reverse pupillary block [10,27,28]. In our cohort, we found that PI did not block the inflow of vitreous humor and the outflow of aqueous humor through the PI. The flow of fluid through the PI was expected to alter the amount of decrease in the height of the nasal sector of the iris and increase the height of the temporal sector at 0 s, but the differences were not significant. However, the ACD was shallower in the ISF group with PI at the baseline. There was one eye with an impending pupillary block that had a history of occasional pupillary blockage and spontaneous recovery in which the iris retracted posteriorly and contacted the anterior surface of IOL even at the stationary baseline position in an eye after ISF without PI (Figure 6). This case was excluded from the analyses, but the iris retracted excessively, and the ACD was deep with the angle widely opened, as described in eyes with reverse pupillary block [12]. These cases are believed to be due to an excessive inflow of vitreous humor into the anterior chamber caused by eye movements. Such cases were not seen in the ISF group with PI. To repair the excess angle recession due to the iris retraction, suturing the iris has been described to prevent pupillary capture by the IOL [9]. 

The results indicated that the PI did not reduce iris fluttering due to eye movements; however, it is assumed the PI prevented a reverse pupillary block by blocking a homeostatic backward movement of the iris. We believe that a simulation of fluid dynamics can be reproduced using a computational model. However, in vivo and live evaluations by AS-OCT are probably more important because the conditions can vary in each case.

Our study has several limitations. First, the number of patients was too few to perform meaningful statistical analyses. In addition, the types of patients in the two groups varied because of their retrospective design. Second, the AS-OCT analyses were performed one month after the surgery and lacked data for longer postoperative follow-up times. Third, we evaluated the iris height with AS-OCT, and we did not evaluate the intracameral flow directly.

## 5. Conclusions

In conclusion, iris fluttering can be detected and quantified by examining consecutive swept-source AS-OCT images. Iris fluttering can act as a check valve for aqueous and vitreous humor movements. The greater amount of iris fluttering by eye movements in eyes after intrascleral fixation of the IOL may cause reverse pupillary block because of the absence of a lens capsular support behind the iris. A PI does not reduce the fluttering of the iris caused by eye movements, but it stabilizes the iris height to prevent a pupillary block.

## Figures and Tables

**Figure 1 jcm-11-04596-f001:**
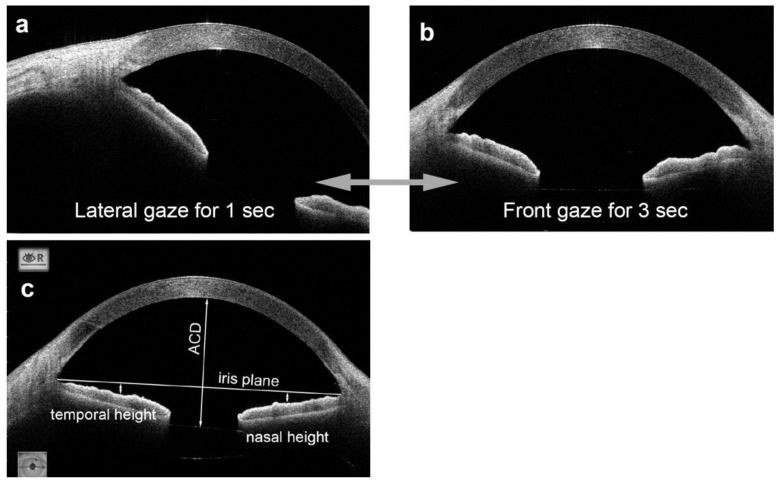
Anterior segment optical coherence tomographic (AS-OCT) images are taken when the eye is fixed on a temporal target and then moves to the primary position. Also shown are images that show the parameters of the AS-OCT images that are measured. (**a**). AS-OCT image when the patient is looking at a point located 30 degrees temporal from the visual axis. (**b**). AS-OCT image taken immediately after the patient has moved the eye to the primary position and holds this position for 3 s. (**c**). The heights of the temporal and nasal sectors of the iris are defined as the distance from the iris plane (the line between anterior chamber angles) to the anterior surface of the nasal and temporal sectors of the mid-iris.

**Figure 2 jcm-11-04596-f002:**
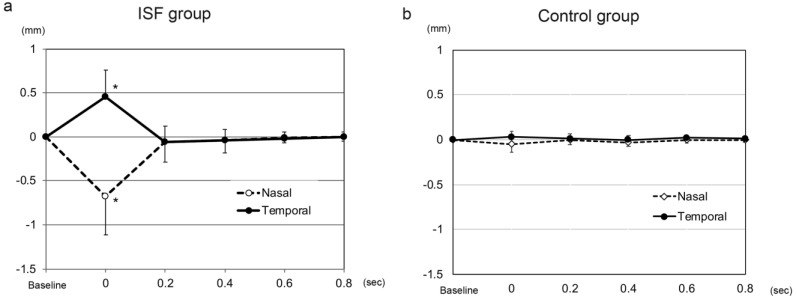
A plot of changes in the height of the temporal and nasal sectors of the iris immediately after a movement of the eye from a 30° temporal position to the primary position in an eye with an intrascleral fixation (ISF) of an intraocular lens (IOL). Also plotted are the values of an eye in which the IOL was implanted in the lens capsule, the control group. (**a**). The height of the nasal sector of the iris of eyes in the ISF group decreases significantly from the baseline at 0 s (*p* < 0.001) and returns at 0.2 s. The height of the temporal sector in eyes of the ISF group increases significantly from the baseline at 0 s and returns to the baseline at 0.2 s. (**b**). The height of the nasal sector of the iris in the ISF group at 0 s was significantly smaller than that of the control group (*p* < 0.001). The temporal height of the ISF group at 0 s was greater than that of the control (*p* < 0.001). The nasal height of the ISF group at 0 s was smaller than that of the control (*p* < 0.001). (* *p* < 0.01).

**Figure 3 jcm-11-04596-f003:**
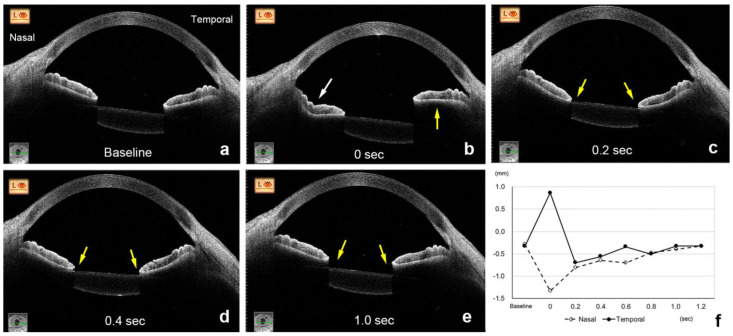
AS-OCT images of a 72-year-old woman in the ISF group. The nasal height of the iris (white arrow) decreases, and the temporal height (yellow arrow) increases at 0 s (**b**) from the baseline (**a**). The height of the nasal and temporal iris (yellow arrows) remains at the lower level at 0.2 s (**c**) and 0.4 s (**d**) and returns to the baseline level at 1.0 s (**e**,**f**).

**Figure 4 jcm-11-04596-f004:**
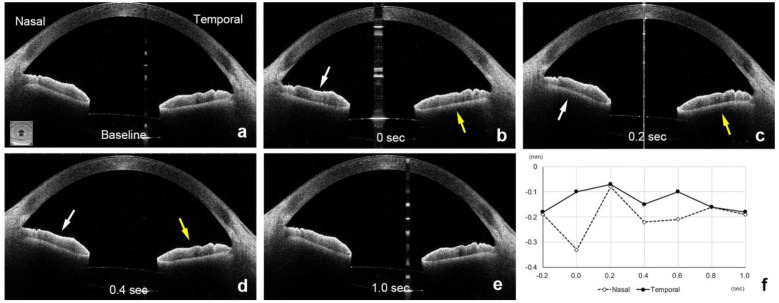
AS-OCT images of a 67-year-old man in the control group who had undergone vitrectomy combined with cataract surgery for an idiopathic macular hole and had the IOL implanted in the capsular bag. The nasal height of the iris (white arrow) decreases at time 0 s (**b**) immediately after the eye stops at the primary position from the baseline (**a**). The nasal height (white arrows) increases at 0.2 s (**c**) and decreases (white arrows) at 0.4 s (**d**). The nasal height returns at 1.0 s (**e**) to the baseline position (**f**). The temporal height of the iris (yellow arrow) increases at 0 s and remains higher at 0.2 s (yellow arrow, **c**) and then returns to the baseline position at 0.4 s (**d**–**f**).

**Figure 5 jcm-11-04596-f005:**
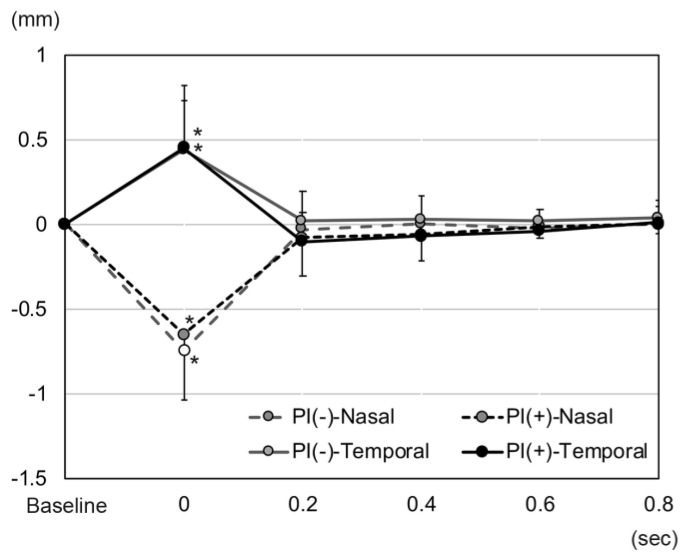
Changes in the height of the iris in the eyes after intrascleral fixation with and without peripheral iridectomy (PI). The nasal height of the iris in eyes with and without PI is significantly lower than the baseline position at 0 s and returned to the baseline position at 0.2 s (* *p* < 0.01). The temporal height in eyes with and without PI was significantly higher than the baseline position and also returned to the baseline position at 0.2 s. The nasal and temporal height of the iris in eyes with PI was not significantly different from that without PI at any time point.

**Figure 6 jcm-11-04596-f006:**
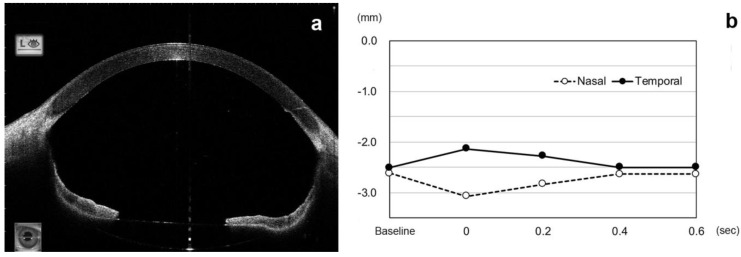
AS-OCT images of a 55-year-old man with an impending pupillary block after intrascleral fixation of an IOL without a peripheral iridectomy. (**a**). AS-OCT image indicates that there is an excessive iris retraction and the anterior chamber depth is 7.07 mm. (**b**). Analysis of the AS-OCT images indicates that the nasal and temporal heights of the iris changes induced by the eye movements, but the height is maintained very low.

**Table 1 jcm-11-04596-t001:** Baseline characteristics between the eyes after intrascleral fixation and vitrectomy combined with cataract surgery.

	ISF	Control	*p*-Value
Eyes	29	15	
Age	66.2 ± 17.7	65.1 ± 8.0	0.435 *
Sex (M/W)	22/7	7/8	0.056 **
Laterality (R/L)	14/15	9/6	0.338 **
Axial length (mm)	25.32 ± 1.73	24.75 ± 2.05	0.202 *
Nasal height of iris (mm)	−0.20 ± 0.16	−0.06 ± 0.17	0.007 *
Temporal height of iris (mm)	−0.23 ± 0.13	−0.08 ± 0.16	0.006 *
ACD (mm)	5.82 ± 0.58	5.25 ± 0.31	<0.001 *
Peripheral iridectomy	20	0	

ISF = intrascleral fixation, M = man, W = woman, R = right, L = left, ACD = anterior chamber depth, * = Mann–Whitney U test, ** = Fisher’s exact probability test.

**Table 2 jcm-11-04596-t002:** Baseline characteristics between the eyes after intrascleral fixation with or without peripheral iridectomy.

	PI (+)	PI (−)	*p*-Value
Eyes	20	9	-
Age	66.9 ± 18.9	64.6 ± 15.4	0.594 *
Sex (M/W)	15/5	7/2	0.631 **
Laterality (R/L)	11/9	3/6	0.250 **
Axial length (mm)	25.52 ± 1.82	24.87 ± 1.48	0.390 *
Nasal height of iris (mm)	−0.24 ± 0.13	−0.14 ± 0.20	0.317 *
Temporal height of iris (mm)	−0.25 ± 0.11	−0.17 ± 0.15	0.116 *
ACD (mm)	5.67 ± 0.46	6.17 ± 0.69	0.030 *

PI = peripheral iridectomy, M = man, W = female, ACD = anterior chamber depth, * = Mann–Whitney test, ** = Fisher’s exact probability test.

## Data Availability

The data presented in this study are available on request from the corresponding author (M.I.).

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
