# Peer review of "Large Amplitude Iris Fluttering Detected by Consecutive Anterior Segment Optical Coherence Tomography Images in Eyes with Intrascleral Fixation of an Intraocular Lens"

_jcm, 2022, doi:10.3390/jcm11154596_

Round 1

Reviewer 1 Report

The authors evaluate the iris fluttering eye movement by AS-OCT in eyes with PPV and ISF versus PPV with IOL and they found that iris fluttering can be quantified by consecutive AS OCT and large amplitude iris fluttering can lead to reverse pupillary block.

In this manuscript the topic could be of interest of ophthalmologist and retina surgeon’s specialist.

The results, findings and conclusion were the main flaw of this research,

The title was very unreadable and do not give much information about the final findings achieve.

The abstract was short and with low information about the results. A purpose was missing. The conclusion was very weak .

The use of AS OCT was previously used to quantify the fluttering of the iris in many previous papers, and it do not suppose a clinical relevance finding to include in a new publication.

The sample size was extremely limited

In addition, some references with similar and related topic were provided in order to justify the lack of necessity of this article.

1.        Ye, S.; Bao, C.; Chen, Y.; Shen, M.; Lu, F.; Zhang, S.; Zhu, D. Identification of Peripheral Anterior Synechia by Corneal Deformation Using  Air-Puff Dynamic Anterior Segment Optical Coherence Tomography. Front. Bioeng. Biotechnol. 2022, 10, 856531, doi:10.3389/fbioe.2022.856531.

2.        Yu, B.; Wang, K.; Zhang, X.; Xing, X. Biometric indicators of anterior segment parameters before and after laser  peripheral iridotomy by swept-source optical coherent tomography. BMC Ophthalmol. 2022, 22, 222, doi:10.1186/s12886-022-02448-1.

3.        Panda, S.K.; Tan, R.K.Y.; Tun, T.A.; Buist, M.L.; Nongpiur, M.; Baskaran, M.; Aung, T.; Girard, M.J.A. Changes in Iris Stiffness and Permeability in Primary Angle Closure Glaucoma. Invest. Ophthalmol. Vis. Sci. 2021, 62, 29, doi:10.1167/iovs.62.13.29.

4.        Shang, Q.; Zhao, Y.; Chen, Z.; Hao, H.; Li, F.; Zhang, X.; Liu, J. Automated Iris Segmentation from Anterior Segment OCT Images with Occludable  Angles via Local Phase Tensor. Annu. Int. Conf. IEEE Eng. Med. Biol.  Soc. IEEE Eng. Med. Biol. Soc. Annu. Int. Conf. 2019, 2019, 4745–4749, doi:10.1109/EMBC.2019.8857336.

5.        Zarei, M.; Mahmoudi, T.; Riazi-Esfahani, H.; Mousavi, B.; Ebrahimiadib, N.; Yaseri, M.; Khalili Pour, E.; Arabalibeik, H. Automated measurement of iris surface smoothness using anterior segment optical  coherence tomography. Sci. Rep. 2021, 11, 8505, doi:10.1038/s41598-021-87954-w.

6.        Ikegawa, W.; Suzuki, T.; Namiguchi, K.; Mizoue, S.; Shiraishi, A.; Ohashi, Y. Changes in Anterior Segment Morphology of Iris Bombe before and after Laser  Peripheral Iridotomy in Patients with Uveitic Secondary Glaucoma. J. Ophthalmol. 2016, 2016, 8496201, doi:10.1155/2016/8496201.

7.        Khodabande, A.; Mohammadi, M.; Riazi-Esfahani, H.; Karami, S.; Mirghorbani, M.; Modjtahedi, B.S. Changes in anterior segment optical coherence tomography following pars plana  vitrectomy without tamponade. Int. J. Retin. Vitr. 2021, 7, 15, doi:10.1186/s40942-021-00285-w.

8.         Zarei, M.; KhaliliPour, E.; Ebrahimiadib, N.; Riazi-Esfahani, H. Quantitative Analysis of the Iris Surface Smoothness by Anterior Segment Optical  Coherence Tomography in Fuchs Uveitis. Ocul. Immunol. Inflamm. 2020, 1–6, doi:10.1080/09273948.2020.1823424.

Author Response

Answers to Comments of Reviewer 1:

We thank you for the comments of our manuscript. The manuscript has been revised accordingly. Our answers are presented below.

The authors evaluate the iris fluttering eye movement by AS-OCT in eyes with PPV and ISF versus PPV with IOL and they found that iris fluttering can be quantified by consecutive AS OCT and large amplitude iris fluttering can lead to reverse pupillary block. In this manuscript the topic could be of interest of ophthalmologist and retina surgeon’s specialist. The results, findings and conclusion were the main flaw of this research.

  1. The title was very unreadable and do not give much information about the final findings achieve. 

Answer: The title was changed to “Large amplitude iris fluttering detected by anterior segment optical coherence tomography in eyes with intrascleral fixation of intraocular lens”

  1. The abstract was short and with low information about the results. A purpose was missing. The conclusion was very weak. 

Answer: the following sentences were changed in line 9, to

“Saccadic eye movements induce movements of the aqueous and vitreous humor and iris fluttering. The iris fluttering during eye movements was evaluated by anterior segment optical coherence tomography (AS-OCT) in 29 eyes with pars plana vitrectomy (PPV) and intrascleral fixation of an intraocular lens (ISF group) and 15 eyes with PPV and an IOL implantation into lens capsular bag (control group).”

“Iris fluttering can act as a check valve for the aqueous and vitreous humor movements and can be detected and quantified by consecutive AS-OCT images. The large amplitude iris fluttering in eyes with intrascleral fixation is important because it can lead to reverse pupillary block.” In line 19.

  1. The use of AS OCT was previously used to quantify the fluttering of the iris in many previous papers, and it do not suppose a clinical relevance finding to include in a new publication. 

Answer: I believe that AS-OCT was used to evaluate the condition of the anterior segment but not for the evaluation of iris fluttering. I have searched in the PubMed and the Google Scholar and there were no hits for “iris fluttering anterior segment optical coherence tomography”.

  1. The sample size was extremely limited

Answer: I agree with your comment. I have described this in the limitation section as follows,

“First, the number of the patients was too few to perform meaningful statistical analyses.” in line 384.

  1. In addition, some references with similar and related topic were provided in order to justify the lack of necessity of this article. 
  2. Ye, S.; Bao, C.; Chen, Y.; Shen, M.; Lu, F.; Zhang, S.; Zhu, D. Identification of Peripheral Anterior Synechia by Corneal Deformation Using Air-Puff Dynamic Anterior Segment Optical Coherence Tomography.  Bioeng. Biotechnol.202210, 856531, doi:10.3389/fbioe.2022.856531.
  3. Yu, B.; Wang, K.; Zhang, X.; Xing, X. Biometric indicators of anterior segment parameters before and after laser peripheral iridotomy by swept-source optical coherent tomography. BMC Ophthalmol.202222, 222, doi:10.1186/s12886-022-02448-1.
  4. Panda, S.K.; Tan, R.K.Y.; Tun, T.A.; Buist, M.L.; Nongpiur, M.; Baskaran, M.; Aung, T.; Girard, M.J.A. Changes in Iris Stiffness and Permeability in Primary Angle Closure Glaucoma.  Ophthalmol. Vis. Sci.202162, 29, doi:10.1167/iovs.62.13.29.
  5. Shang, Q.; Zhao, Y.; Chen, Z.; Hao, H.; Li, F.; Zhang, X.; Liu, J. Automated Iris Segmentation from Anterior Segment OCT Images with Occludable Angles via Local Phase Tensor.  Int. Conf. IEEE Eng. Med. Biol.  Soc. IEEE Eng. Med. Biol. Soc. Annu. Int. Conf.20192019, 4745–4749, doi:10.1109/EMBC.2019.8857336.
  6. Zarei, M.; Mahmoudi, T.; Riazi-Esfahani, H.; Mousavi, B.; Ebrahimiadib, N.; Yaseri, M.; Khalili Pour, E.; Arabalibeik, H. Automated measurement of iris surface smoothness using anterior segment optical coherence tomography.  Rep.202111, 8505, doi:10.1038/s41598-021-87954-w.
  7. Ikegawa, W.; Suzuki, T.; Namiguchi, K.; Mizoue, S.; Shiraishi, A.; Ohashi, Y. Changes in Anterior Segment Morphology of Iris Bombe before and after Laser Peripheral Iridotomy in Patients with Uveitic Secondary Glaucoma.  Ophthalmol.20162016, 8496201, doi:10.1155/2016/8496201.
  8. Khodabande, A.; Mohammadi, M.; Riazi-Esfahani, H.; Karami, S.; Mirghorbani, M.; Modjtahedi, B.S. Changes in anterior segment optical coherence tomography following pars plana vitrectomy without tamponade.  J. Retin. Vitr.20217, 15, doi:10.1186/s40942-021-00285-w.
  9. Zarei, M.; KhaliliPour, E.; Ebrahimiadib, N.; Riazi-Esfahani, H. Quantitative Analysis of the Iris Surface Smoothness by Anterior Segment Optical Coherence Tomography in Fuchs Uveitis.  Immunol. Inflamm.2020, 1–6, doi:10.1080/09273948.2020.1823424.

Answer: Thank you for your suggestion. The following sentences were changed in line 46 with the addition of the references.

“The anterior segment optical coherence tomographic (AS-OCT) devices were designed to record cross-sectional images of the anterior segment of the eye [18,19].

The smoothness of the anterior surface of the iris in Fuchs uveitis [20,21], morphological changes after laser peripheral iridectomy in eyes with primary angle closure glaucoma and uveitic secondary glaucoma [22,23], iris movement during pupil constriction with light in primary angle closure glaucoma [24], identification of peripheral anterior synechia by corneal deformation using air-puff [25], have been evaluated with AS-OCT. Swept source AS-OCT uses a longer wavelength light source of 1310 nm which enables the scanning of the anterior and posterior surface of an IOL. It also enables high-speed scanning leading to images of the entire anterior chamber angle, and the repetitive imaging of the same section of the anterior chamber will appear similar to a continuous movie scan [18,19].”

  1. Zarei, M.; KhaliliPour, E.; Ebrahimiadib, N.; Riazi-Esfahani, H. Quantitative Analysis of the Iris Surface Smoothness by Anterior Segment Optical Coherence Tomography in Fuchs Uveitis.  Immunol. Inflamm.2020, 1–6, doi:10.1080/09273948.2020.1823424.
  2. Zarei, M.; Mahmoudi, T.; Riazi-Esfahani, H.; Mousavi, B.; Ebrahimiadib, N.; Yaseri, M.; Khalili Pour, E.; Arabalibeik, H. Automated measurement of iris surface smoothness using anterior segment optical coherence tomography.  Rep.202111, 8505, doi:10.1038/s41598-021-87954-w.
  3. Yu, B.; Wang, K.; Zhang, X.; Xing, X. Biometric indicators of anterior segment parameters before and after laser peripheral iridotomy by swept-source optical coherent tomography.  Ophthalmol.202222, 222, doi:10.1186/s12886-022-02448-1.
  4. Ikegawa, W.; Suzuki, T.; Namiguchi, K.; Mizoue, S.; Shiraishi, A.; Ohashi, Y. Changes in Anterior Segment Morphology of Iris Bombe before and after Laser Peripheral Iridotomy in Patients with Uveitic Secondary Glaucoma.  Ophthalmol.20162016, 8496201, doi:10.1155/2016/8496201.
  5. Panda, S.K.; Tan, R.K.Y.; Tun, T.A.; Buist, M.L.; Nongpiur, M.; Baskaran, M.; Aung, T.; Girard, M.J.A. Changes in Iris Stiffness and Permeability in Primary Angle Closure Glaucoma.  Ophthalmol. Vis. Sci.202162, 29, doi:10.1167/iovs.62.13.29.
  6. Ye, S.; Bao, C.; Chen, Y.; Shen, M.; Lu, F.; Zhang, S.; Zhu, D. Identification of Peripheral Anterior Synechia by Corneal Deformation Using Air-Puff Dynamic Anterior Segment Optical Coherence Tomography.  Bioeng. Biotechnol.202210, 856531, doi:10.3389/fbioe.2022.856531.

Reviewer 2 Report

Authors discuss the iris fluttering during eye movements  by means of anterior segment optical coherence tomography in eyes with pars plana vitrectomy and intrascleral fixation of an intraocular lens  versus  15 eyes with PPV and an IOL implantation into lens capsular 11 bag (control group). English is accurate and conclusion are in sound with the well written and interesting study. I would recommend publication after a minor revision to the following queries:

Please discuss more accurately the following sentence “Because there is no lens 26 capsule behind the iris, a fluttering of the iris can be seen through the slit-lamp microscope 27 when the patient blinks” page 1 line 26. Actually iris fluttering may be seen in phakic eye as well.

Page 1 line 34: please also add the so called UGH syndrome

Please provide evidence and reference of the following sentence : “The aqueous humor is more susceptible to the inertia moments than the vitreous humor during eye movements  because the aqueous humor is located further from the center of the eye” line 324 page 10. Are the Authors sure for this affirmation? Actually the vitreous is gelatinous while the acqueous humor is a watery fluid. So the inert passive angular movement should be higher for vitreous gel, please discuss it more accurately.

Author Response

Answers to Comments to Reviewer 2:

We thank you for the comments of our manuscript. The manuscript has been revised accordingly, and our answers are included.

Authors discuss the iris fluttering during eye movements by means of anterior segment optical coherence tomography in eyes with pars plana vitrectomy and intrascleral fixation of an intraocular lens versus 15 eyes with PPV and an IOL implantation into lens capsular bag (control group). English is accurate and conclusion are in sound with the well written and interesting study. I would recommend publication after a minor revision to the following queries:

1.Please discuss more accurately the following sentence “Because there is no lens capsule behind the iris, a fluttering of the iris can be seen through the slit-lamp microscope when the patient blinks” page 1 line 26. Actually iris fluttering may be seen in phakic eye as well.

Answer: The sentence was changed to,

“A fluttering of the iris can be seen more frequently in eyes without a lens capsule behind the iris through the slit-lamp microscope when the patient blinks”.

2.Page 1 line 34: please also add the so called UGH syndrome

Answer: The sentence was changed to,

“Pupillary capture by the IOL can cause an increase of the intraocular pressure and mild eye pain and discomfort, the so called Uveitis-Glaucoma-Hyphema (UGH) syndrome.”

3.Please provide evidence and reference of the following sentence : “The aqueous humor is more susceptible to the inertia moments than the vitreous humor during eye movements  because the aqueous humor is located further from the center of the eye” line 324 page 10. Are the Authors sure for this affirmation? Actually the vitreous is gelatinous while the acqueous humor is a watery fluid. So the inert passive angular movement should be higher for vitreous gel, please discuss it more accurately.

Answer: We thank you for this comment. I understand that the passive angular movement are different between the vitreous gel and fluid. Both groups consisted of vitrectomized eyes and the following sentence was revised to,

“The aqueous humor is more susceptible to the inertia moments than the vitreous fluid in vitrectomized eyesduring eye movements because the aqueous humor is located further from the center of the eye.”

Round 2

Reviewer 1 Report

Thanks for the comment, in this case the manuscript does not solved the comment

In this case, I could reccomend this research to continue with the publication process

Author Response

Answers to Comments of Reviewer 1:

We thank you for the comments on our manuscript. The manuscript has been revised accordingly. Our answers are presented below.

Thanks for the comment, in this case the manuscript does not solved the comment

In this case, I could reccomend this research to continue with the publication process

Answer: We thank you for the comments. The manuscript has been made further modifications based on the previous comments.

  1. The title was very unreadable and do not give much information about the final findings achieve. 

Answer: The title was changed to “Large amplitude iris fluttering detected by consecutive anterior segment optical coherence tomography images in eyes with intrascleral fixation of an intraocular lens”

The abstract was short and with low information about the results. A purpose was missing. The conclusion was very weak. 

Answer: The following sentences were changed in line 10, to “To evaluate the iris fluttering during eye movements, anterior segment optical coherence tomography (AS-OCT) was used in 29 eyes with pars plana vitrectomy (PPV) and intrascleral fixation of an intraocular lens (ISF group) and 15 eyes with PPV and an IOL implantation into lens capsular bag (control group).

  1. The use of AS-OCT was previously used to quantify the fluttering of the iris in many previous papers, and it do not suppose a clinical relevance finding to include in a new publication. 

Answer: The following sentences were changed in line 10. To “The smoothness of the anterior surface of the iris in Fuchs uveitis [20,21], morphological change after laser peripheral iridectomy in eyes with primary angle closure glaucoma and uveitic secondary glaucoma have been evaluated by AS-OCT [22,23]. Swept-source AS-OCT uses a longer wavelength light source of 1310 nm which enables the scanning of the anterior and posterior surface of an IOL. It also enables high-speed scanning leading to images of the entire anterior chamber angle, and the repetitive imaging of the same section of the anterior chamber will appear similar to a continuous movie scan [18,19]. With the dynamic scanning ability of swept-source AS-OCT, evaluation of iris movement during pupil constriction with light revealed that the iris stiffness was higher in patients with primary angle-closure glaucoma than in healthy controls [24]. Detection of peripheral anterior synechia has been described by corneal deformation using an air-puff dynamic AS-OCT system [25].

We present a dynamic analysis with swept-source AS-OCT which allowed us to evaluate and quantify the degree of iris fluttering produced by eye movements in eyes with an intrasclerally fixed IOL.

4.Does the introduction provide sufficient background and include all relevant references? (Must be improved)

Answer: The section of the instruction has been revised as above (No.3) with adding the description of swept-source AS-OCT.

  1. Are all the cited references relevant to the research? (Must be improved)

Answer: The following sentences were changed in line 329 with the addition of the references, “In this study, all eyes were vitrectomized. However, it has been described that vitrectomy without gas tamponade is not associated with changes in anterior chamber morphology evaluated with AS-OCT [26].”

  1. Khodabande, A.; Mohammadi, M.; Riazi-Esfahani, H.; Karami, S.; Mirghorbani, M.; Modjtahedi, B.S. Changes in anterior segment optical coherence tomography following pars plana vitrectomy without tamponade. Int. J. Retin. Vitr.20217, 15, doi:10.1186/s40942-021-00285-w.

  1. Is the research design appropriate? (Must be improved)

Answer: I believe the research design of this case-control study is appropriate even though the numbers of enrolled patients are small.

  1. Are the methods adequately described? (Must be improved)

Answer: Figure 1 has been revised.

  1. Are the results clearly presented? (Must be improved)

Answer: The following sentences were changed in line 344, “When the nasal sector of the iris is pushed posteriorly, the pupil margins contact the anterior surface of the IOL, and the flow of the aqueous humor stops as if a valve was closed (attachment of the pupil margin to the IOL). On the other hand, even if the temporal sector of the iris is lifted due to the inflow of vitreous humor into the anterior chamber, the pupil margin does not contact the IOLs. Thus, the flow of the vitreous humor into the anterior chamber on the temporal side is enhanced more than the outflow of aqueous humor on the nasal side as the valve was opened.”

  1. Are the conclusions supported by the results? (Must be improved)

Answer: The following sentences were changed in line 398, “In conclusion, iris fluttering can be detected and quantified by examining consecutive swept-source AS-OCT images. Iris fluttering can act as a check valve for aqueous and vitreous humor movements. The greater amount of iris fluttering by eye movements in eyes after intrascleral fixation of the IOL may cause reverse pupillary block because of the absence of a lens capsular support behind the iris.”
